# The single-cell transcriptomic atlas and RORA-mediated 3D epigenomic remodeling in driving corneal epithelial differentiation

Mingsen Li [1,4] ✉, Huizhen Guo[1,4], Bofeng Wang[1,4], Zhuo Han[1], Siqi Wu[1], Jiafeng Liu[1], Huaxing Huang[1], Jin Zhu[1], Fengjiao An[1], Zesong Lin[1], Kunlun Mo[1], Jieying Tan[1], Chunqiao Liu[1], Li Wang[1], Xin Deng [2], Guigang Li[3], Jianping Ji [1] ✉ & Hong Ouyang [1] ✉

Proper differentiation of corneal epithelial cells (CECs) from limbal stem/progenitor cells (LSCs) is required for maintenance of ocular homeostasis and clear vision. Here, using a single-cell transcriptomic atlas, we delineate the comprehensive and refined molecular regulatory dynamics during human CEC development and differentiation. We find that RORA is a CEC-specific molecular switch that initiates and drives LSCs to differentiate into mature CECs by activating PITX1. RORA dictates CEC differentiation by establishing CEC-specific enhancers and chromatin interactions between CEC gene promoters and distal regulatory elements. Conversely, RORA silences LSC-specific promoters and disrupts promoter-anchored chromatin loops to turn off LSC genes. Collectively, our work provides detailed and comprehensive insights into the transcriptional dynamics and RORA-mediated epigenetic remodeling underlying human corneal epithelial differentiation.

The limbal and corneal epithelia lie adjacent to each other, and are stratified, non-keratinized squamous epithelia that cover the anterior corneal surface. As protective barriers, they are composed of basal, suprabasal and superficial layers, and play an essential role in maintaining the structural and functional integrity of the cornea[1,2]. Limbal stem/progenitor cells (LSCs) residing in the basal layer of the limbus are responsible for the continuous replenishment of the corneal epithelium[3]. LSCs asymmetrically divide and produce both stem cells and transit-amplifying cells that further proliferate, migrate upward and centripetally, and eventually differentiate into three specialized post-mitotic corneal epithelium cell (CEC) types: basal, suprabasal and superficial cells[4,5]. Aberrant corneal epithelial differentiation is associated with some common corneal diseases, such as corneal perforation, keratitis and keratohelcosis[6]. Despite the significance of corneal epithelial development and homeostasis, cellular diversity, molecular

characteristics and development and differentiation trajectories of the corneal epithelium have not been well-documented.

The balance between self-renewal and differentiation of LSCs is tightly controlled to ensure efficient corneal epithelial turnover and the maintenance of corneal homeostasis and transparency. The well-known LSC markers TP63[3], C/EBPD[7], BCAM[8], IFITM3[9] and ABCB5[10] are key regulators that maintain LSC function and self-renewal. Although KLF4[11,12], PAX6[13,14], EHF[12] and FOXC1[15] are known to promote corneal epithelial differentiation, they are not CEC-specific and are also expressed in basal, suprabasal and superficial layers of the limbal epithelium. The CEC-specific molecular switch that can initiate and dictate LSC differentiation program remains largely unknown.

In this study, we used single-cell RNA-sequencing (scRNA-seq) data to depict detailed molecular atlases containing gene co-expression network, SCENIC transcription factor (TF) activity, RNA

[1]State Key Laboratory of Ophthalmology, Zhongshan Ophthalmic Center, Sun Yat-sen University, Guangdong Provincial Key Laboratory of Ophthalmology and Visual Science, Guangdong Provincial Clinical Research Center for Ocular Diseases, Guangzhou 510060, China. [2]Department of Biomedical Sciences, City University of Hong Kong, Hong Kong 999077, China. [3]Department of Ophthalmology, Tongji Hospital, Tongji Medical College, Huazhong University of Science and Technology, Wuhan, Hubei Province, China. [4]These authors contributed equally: Mingsen Li, Huizhen Guo, Bofeng Wang. ✉e-mail: lims3@mail2.sysu.edu.cn; jpji1974@126.com; Ouyhong3@mail.sysu.edu.cn

velocity, gene expression kinetics, pseudotime inference and cell state transition along the development and differentiation trajectories of human corneal epithelium. We found that RORA, as a molecular switch, was able to induce LSCs to differentiate into mature CECs via PITX1. Depletion of the RORA–PITX1 axis markedly blocked CEC differentiation. RORA established lineage-specific promoter-anchored chromatin interaction and epigenetic landscapes that dictated CEC differentiation. Our work delineated the transcriptional dynamics and RORA-shaped epigenomic landscape and three-dimensional genome architecture in driving CEC differentiation.

## Results

### Single-cell atlas reveals molecular trajectories of human corneal epithelial development and differentiation

Although many single-cell transcriptomic datasets of human ocular surface tissues have been generated in previous studies[16–18], a comprehensive and detailed molecular dynamic profile underlying corneal epithelial development and differentiation has not yet been well characterized. We collected and merged a set of published single-cell transcriptomic datasets[16], including 133,490 high-quality single cells from healthy human corneal–conjunctival tissues at 10–21 post-conception week (PCW) and in adulthood. Unsupervised clustering of these cells identified 17 distinct cell types representing the ocular epithelium, endothelium, keratocytes, fibroblasts, melanocytes and immune cell populations annotated with canonical lineage markers (Supplementary Fig. 1a–c). We further clustered the limbal–corneal epithelial compartments, identifying 10 discrete cell subpopulations corresponding to LSCs and the basal, suprabasal and superficial layers of the embryonic and adult limbus and cornea (Fig. 1a). Unbiased potential of heat-diffusion for affinity-based transition embedding (PHATE)[19] and latent-temporal estimation[20] inferred the expected lineage trajectory from embryo to adult in the order of LSCs, basal, suprabasal and superficial layers (Fig. 1b). Differential gene expression analysis revealed unique signature genes in each cell type (Supplementary Fig. 1d). For example, as well as the LSC markers *KRT15* and *KRT14*, adult (but not embryonic) LSCs specifically expressed *S100A2* and *IFITM3*[9], which was further confirmed by immunofluorescence staining (Fig. 1c, d and Supplementary Fig. 1d). S100A2 was located at the basal layer of the limbus and more than half of the S100A2-positive cells were KRT15-postive. IFITM3 was also co-located with LSC marker KRT19 in the basal layer of the limbus (Fig. 1d). We found that *MYC*[21], *TP63*[1], *PEDF*[22] and *CEBPD*[7], which are associated with LSC proliferation, were highly expressed in adult LSCs and their expression dramatically decreased with the differentiation trajectory (Supplementary Fig. 1e). In addition, *TP63* and *PEDF* were also expressed in embryonic LSCs (Supplementary Fig. 1e). The superficial layers of the adult limbus and cornea shared many markers, including the newly identified *LGALS3* (Fig. 1c and Supplementary Fig. 1d), which contributes to corneal repair[23]. Consistent with KRT12 and CLU, LGALS3 was primarily expressed in the suprabasal and superficial layers of the limbus and cornea from embryo to adulthood (Fig. 1c, d).

High dimensional weighted gene co-expression network analysis (hdWGCNA)[24] defined six gene co-expression modules and constructed a cell-type-specific gene interaction network for each module (Fig. 1e, f and Supplementary Fig. 1f, g). Terminal differentiation-associated modules 1 and 2, containing hub genes *LGALS3*, *MAL2* and *S100A4* (Fig. 1f and Supplementary Fig. 1h, i), were enriched for Gene Ontology (GO) terms related to glycoprotein processing, lipid droplet formation, p38 MAPK signaling pathway, tight junction and epidermal cell differentiation (Fig. 1g). Module 3 including the canonical markers (*CLU*, *KRT12* and *ALDH3A1*)[1,25] was specific to the differentiating adult corneal epithelium and was involved in the response to cadmium, copper and zinc ions (Fig. 1g). The less differentiated epithelial layers in adults (module 4) were enriched for GO terms associated with fatty acid and vitamin D metabolism. Embryonic epithelial-specific modules

5 and 6 were linked to collagen fibril organization and metabolism of protein and mRNA. The identification of *PTN*, *CRABP2*, *BCAM*, *GPC3*, *SPARC*, *MEG3*, *WNT6*, *EEF1A1*, *RPL6* and *RPS3* as hub genes in module 6 implied their potential importance in embryonic LSCs (Supplementary Fig. 1g, h, j).

To characterize the developmental trajectory of limbal and corneal epithelia, we performed SCENIC analysis[26] to dissect cell-state-specific TF activity (Fig. 1h). We found that EHF, ELF3, BHLHE40 and MAFF were the general TFs for adult limbal and corneal epithelia. MAZ, BCL11A, IRX3 and ASCL2 were specific to embryonic LSCs, while MYC and the well-documented ETS1[27] were exclusive to adult LSCs. We also observed a cohort of regulators (e.g., KLF13, IRF2, KMT2A and KAT2A) that were active across the entire developmental trajectory.

We then leveraged the machine-learning-dependent Dynamo analytical framework[28] to characterize the corneal epithelial differentiation dynamics by reconstructing continuous transcriptomic vector fields in the full gene-expression-state space. As expectedly, the RNA velocity flow and ddhodge potential (vector field-based pseudotime) recapitulated the natural development and differentiation progression of the corneal epithelium (Fig. 1i). We found that the cells gradually accelerated during development and differentiation, with initial embryonic LSCs having the lowest acceleration and the terminal adult corneal superficial layer having the highest acceleration (Fig. 1j). By contrast, the cell divergence field, which reflects differentiation potential and stem cell plasticity, was high in embryonic and adult LSCs, but significantly lower in differentiated cells (Fig. 1j). We also revealed the gene acceleration kinetics along the developmental pseudotime (Fig. 1k). The high acceleration of *WNT7B*, *WNT10A* and the master regulator *PAX6* in early pseudotime highlighted their potential importance in early LSC fate commitment (Fig. 1k). *KLF4*, *KLF5*[12,29] and *EHF* showed high acceleration in the late stages of differentiation, consistent with their well-known pro-differentiation functions (Fig. 1l).

### RORA and PITX1 were potential regulators for corneal epithelial differentiation in adults

We then sought to identify the core TFs that could induce the transition from LSCs to CECs. We first applied the CellRank modular framework[20], based on Markov state modeling of single-cell data, to reconstruct the molecular dynamics of corneal epithelial differentiation, identifying 21 TFs whose expression trends in pseudotime correlated best with terminal fate probabilities, including the well-documented *KLF4*, *KLF5* and *EHF* (Fig. 2a). Primary adult LSCs were isolated, expanded and induced to differentiate into mature CECs (Fig. 2b and Supplementary Fig. 2a), as described previously[27]. Using the established in vitro differentiation model, we obtained the differentially expressed genes between LSCs and CECs by bulk-RNAseq (Fig. 2c). LSC-specific *ETS1*[27] and *HMGA2*[27] were inhibited and those well-known CEC signatures were activated in the induced CECs (Fig. 2c). We then overlapped the CEC-specific TFs with above CellRank TFs, generating four candidate regulators: RORA, PITX1, DBP and NR1D1 (Fig. 2d).

We then focused on *RORA* and *PITX1*, which were specifically expressed in the differentiated limbal–corneal epithelial layers of adults but not in LSCs (Fig. 2e, f and Supplementary Fig. 2b). RORA exhibited the top 1 SCENIC specificity score in both the suprabasal and superficial layers of the adult corneal epithelium (Fig. 2g, h). PITX1 also showed high and exclusive regulon activity in adult CECs (Fig. 2g, h). In addition, their RNA splicing velocities dramatically increased with the differentiation trajectory (Fig. 2i). Furthermore, k-Nearest-Neighbors Conditional-Density Resampled Estimate of Mutual Information (kNN-DREMI) analysis[30], which quantifies the strength of the relationship between two genes via scRNA-seq data, showed that *RORA* and *PITX1* were positively correlated with the differentiation markers of corneal epithelium in vivo (Fig. 2j). Collectively, these observations raised the possibility that RORA and PITX1 might contribute to corneal epithelial differentiation in adults.

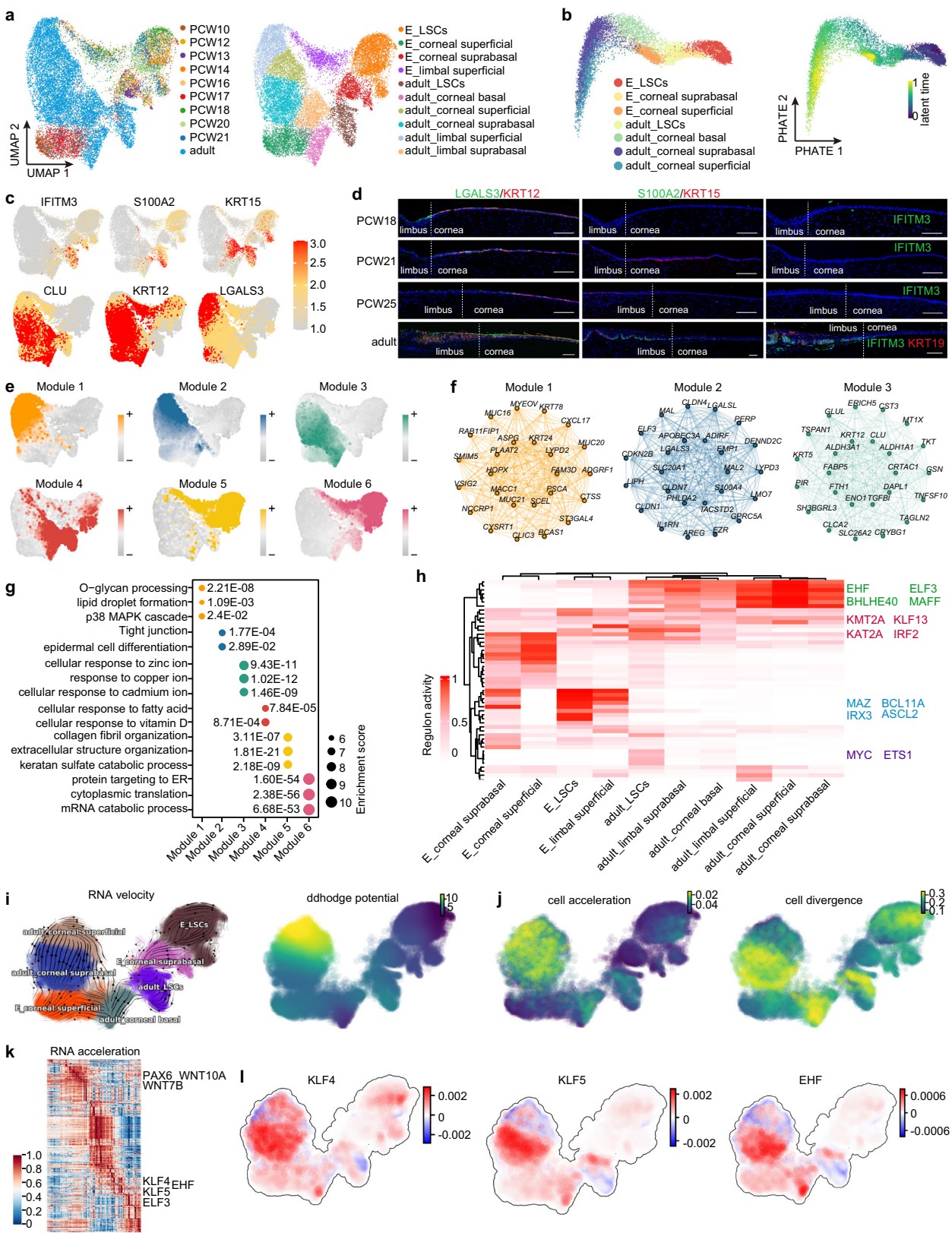

## RORA is a molecular switch that initiates and drives corneal epithelial differentiation via PITX1

Recently, CellOracle[31], a machine learning-based approach, was developed to perform in silico TF perturbations and simulate the consequent cell identity diversions using scRNA-seq datasets. To investigate the in vivo function of *RORA* and *PITX1* in human corneal tissue, we used CellOracle to predict global changes in gene expression following *RORA* or *PITX1* simulation knockout based on our own SCENIC gene regulatory network model inferred from all limbal–corneal epithelial cell types. The normal pseudotime and developmental vector field flow estimated via CellOracle recapitulates the Dynamo results shown in Fig. 1i (Fig. 3a). Notably, the direction of

**Fig. 1 | A single-cell atlas of the limbal−corneal epithelium. a** Uniform Manifold Approximation and Projection (UMAP) plots showing separated cell types of the limbal and corneal epithelia from the indicated human embryonic (E) and adult samples. **b** PHATE maps of the indicated cell types in the embryonic and adult limbal−corneal epithelium (left) and latent time estimation (right). **c** UMAP plots showing the expression of the indicated genes. **d** Immunofluorescence staining of the limbal−corneal tissues at PCW18, PCW21, PCW25 and adulthood for the indicated genes. Scale bar, 100 μm. *n* = 3 biologically independent experiments. **e** UMAP plots of the six gene co-expression modules identified through hdWGCNA analysis. **f** hdWGCNA module network plots showing the top 25 genes by the

eigengene-based connectivity for module 1, 2, 3. Each edge represents the co-expression relationship between two genes in the network. The top 10 hub genes are placed in the center of the plot, while the remaining 15 genes are placed in the outer circle. **g** GO enrichment analysis of each hdWGCNA module. **h** Heatmap showing SCENIC TF activities of the indicated cell types. **i** RNA velocity (left) and ddhodge potential (right) estimation of the indicated cell types by Dynamo. **j** Cell acceleration and divergence estimation by Dynamo. **k** Dynamo gene acceleration kinetic heatmap across vector field based pseudotime. **l** The acceleration for *KLF4*, *KLF5*, and *EHF* estimated by Dynamo.

the CellOracle knockout vector of *RORA* or *PITX1* is opposite to that of the natural differentiation vector in the suprabasal and superficial cell layers (Fig. 3a and Supplementary Fig. 3a), suggesting that *RORA* or *PITX1* knockout blocks corneal epithelial differentiation. Dynamo can perform in silico multiple gene perturbations or activations, and predict cell-state shifts after genetic perturbations. In line with the results of CellOracle, in silico double perturbation of *RORA* and *PITX1* by Dynamo induces backward corneal epithelial differentiation, which is opposite to the normal developmental trajectory shown in Fig. 1i (Fig. 3b). In contrast, double activation of *RORA* and *PITX1* promotes the differentiation progression (Fig. 3b).

RNA Jacobian tensor (cell by gene by gene) calculation for genes across all cells, with the reconstructed vector field by Dynamo, allows us to predict the gene regulation networks. We then performed RNA Jacobian analysis to infer the regulation from *RORA* and *PITX1* to the selected key lineage genes. We found that *RORA* activates the expression of the CEC lineage markers (*KRT3*, *KRT12*, *LGALS3*, *MAL*, *MAL2* and *S100A4*) in vivo (Fig. 3c). The inhibitory effects of *RORA* on LSC stemness genes (*IFITM3*, *KRT15*, *KRT19*, *KRT14*, *BCAM* and *S100A2*) progressively increased with an increase in RNA splicing velocity of *RORA* (Fig. 3c). Similarly, RNA Jacobian analysis predicted that *PITX1* can promote the expression of CEC markers, while simultaneously suppressing the expression of LSC genes (Supplementary Fig. 3b). Both Dynamo and CellOracle in silico perturbation and activation simulations suggested that *RORA* and *PITX1* are required for human corneal epithelial differentiation in vivo.

To verify the predicted in vivo functions of *RORA* and *PITX1*, we overexpressed them separately in isolated primary LSCs and found that their overexpression significantly attenuated LSC proliferative capacity and the expression of *KRT15*, *BCAM*, *IFITM3*, *ETS1*, *HMGA2*[27] and *S100A2* (Fig. 3d, e and Supplementary Fig. 3c), which are related to LSC stemness and proliferation. In contrast, they activated the CEC signatures *KRT3*, *KRT12*, *KRT24*, *CLU*, *LGALS3*, *MAL2* and the pro-differentiation TFs *KLF4* and *PAX6* (Fig. 3e and Supplementary Fig. 3c). Dysfunction of RORA via SR3335[32] (an inverse agonist of RORA) or knockdown of *PITX1* significantly perturbed CEC differentiation, as evidenced by the loss of differentiation markers (Supplementary Fig. 3d, e). Importantly, the terminal differentiation-associated gene co-expression modules identified by scRNA-seq data (shown in Fig. 1e) and the CEC-associated genes identified by in vitro differentiation model (shown in Fig. 2c) were up-regulated in *RORA*- and *PITX1*-over-expressed LSCs, whereas LSC-specific gene sets were significantly suppressed (Fig. 3f, g). Conversely, the loss of RORA or PITX1 significantly inhibited the expression of the CEC-specific gene sets upon differentiation (Supplementary Fig. 3f). These observations highlighted that RORA and PITX1 were able to initiate and dominate CEC differentiation.

Further kNN-DREMI analysis revealed that the expression of *RORA* and *PITX1* was positively correlated in human limbal−corneal epithelial tissue (Fig. 3h). The gene expression changes triggered by *RORA* and *PITX1* overexpression also exhibited a highly positive correlation (Fig. 3i), suggestive of their functional consistency. RNA Jacobian analysis suggested that *RORA* activates itself and *PITX1* in the suprabasal and superficial layers of the human corneal epithelium (Fig. 3j),

which was further verified by in vitro *RORA* overexpression experiment (Fig. 3k). However, *PITX1* did not regulate *RORA* (Fig. 3k). *PITX1* over-expression rescued the perturbation of CEC differentiation that was caused by SR3335 (Fig. 3l). Thus, RORA was a key upstream regulator of PITX1. In a word, we highlighted that RORA-PITX1 axis plays a key role in the corneal epithelial differentiation in vitro.

## RORA dictates CEC differentiation by establishing lineage-specific chromatin interactions and epigenetic architecture

Chromatin Immunoprecipitation Sequencing (ChIP-seq) revealed that RORA and PITX1 co-occupied the active promoters (H3K27ac/H3K4me3-postive and H3K27me3-negative) and enhancers (H3K27ac-postive and H3K4me3/H3K27me3-negative) (Supplementary Fig. 4a). RORA-binding sites were enriched for the motifs of RORA, PITX1 and other pro-differentiation TFs (Supplementary Fig. 4b), suggesting that RORA functioned by binding to TF hotspots that regulated CEC differentiation. We then applied the genome-wide promoter capture Hi-C sequencing[33] to generate high-resolution maps of the interactions between gene promoters and their regulatory elements in control and *RORA*-overexpressed LSCs (Fig. 4a). A set of differential promoter-anchored chromatin loops was identified, consisting of ~20% promoter−promoter (P−P) and ~80% promoter−nonpromoter region (P−NPR) interactions in each group (Fig. 4b). Interaction anchors of the differential loops were rarely shared between the two groups (Supplementary Fig. 4c). RORA primarily bound to the promoter anchors but not to the NPRs of the *RORA*-overexpression-specific loops (Fig. 4c). The *RORA*-overexpression-specific promoter anchors regulated the genes that were involved in epithelial cell differentiation, eye development, lipid biosynthetic process, sterol metabolic process and calcium signaling pathways (Fig. 4d).

To dissect RORA-induced chromatin remodeling, the changes of H3K27ac (active state), H3K27me3 (repressive state) and H3K4me3 (promoter-specific) modifications were compared upon *RORA* over-expression. H3K27ac varied the most, with 7468 increased and 4949 decreased peaks, whereas only 1804 increased and 1426 decreased H3K27me3 peaks were observed (Fig. 4e). RORA primarily diminished H3K4me3 deposition. The RORA-established H3K27ac regions were bound by RORA and were mainly putative enhancers characterized by negative H3K4me3 and H3K27me3 (Fig. 4f). These RORA-induced enhancers contributed to the activation of the CEC gene sets (Supplementary Fig. 4d). In contrast, RORA decommissioned H3K27ac and H3K4me3 deposition only at the promoters of LSC genes (Fig. 4f), resulting in corresponding transcriptional inhibition (Supplementary Fig. 4d). For another subset of H3K4me3-marked promoters, H3K27ac depletion was accompanied by H3K27me3 gain upon *RORA* over-expression (Fig. 4g). This switch from active to bivalent status contributed to transcriptional regression, as evidenced by the *BCAM and IFITM3* loci (Supplementary Fig. 4e). Conversely, a subset of CEC-specific genes became de-repressed by losing H3K27me3 and simultaneously gaining H3K27ac at their promoters and/or enhancers after RORA binding (Fig. 4g).

Remarkably, we found that the coupling of chromatin three-dimensional interactions with histone modifications dominated RORA-mediated gene expression changes. In LSCs, the promoters of *RORA*

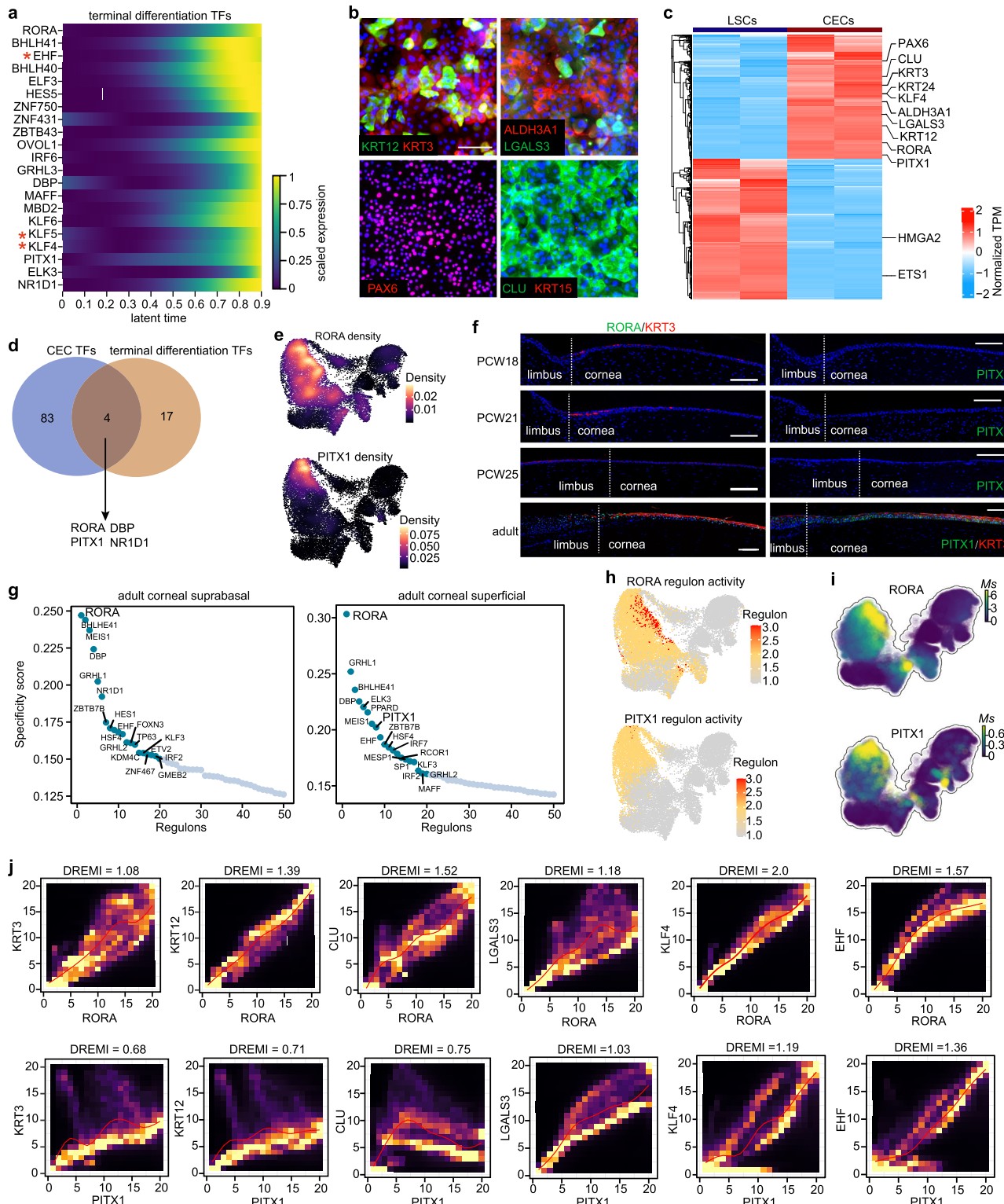

**Fig. 2 | RORA and PITX1 are potential regulators of CEC differentiation.**
**a** Smoothed expression trends in pseudotime of TFs whose expression correlates best with terminal fate probabilities calculated by CellRank. **b** Immunofluorescence staining of the indicated genes in the induced CECs that differentiated for 7 days. Scale bar, 100 μm. *n* = 3 biologically independent experiments. **c** Heatmap showing the differentially expressed genes between cultured LSCs and induced CECs. TPM: transcripts per kilobase million. **d** Overlapping of CellRank TFs (**a**) and CEC-specific TFs (**c**). **e** UMAP plots of *RORA* and *PITX1* density. **f** Immunofluorescence staining of RORA, PITX1 and KRT3 in PCW18, PCW21, PWC25 and adult limbal–corneal tissues. Scale bar, 100 μm. *n* = 3 biologically independent experiments. **g** Ranked SCENIC TFs in adult corneal suprabasal and superficial layers. **h** UMAP plots showing SCENIC regulon activities of RORA and PITX1. **i** UMAP plots showing the scores for moments of spliced RNAs (*Ms*) of *RORA* and *PITX1*. **j** kNN-DREMI analysis (DREVI plots and DREMI values) indicates the relationship between the indicated genes after imputation with MAGIC (Markov affinity-based graph imputation of cells)[30].

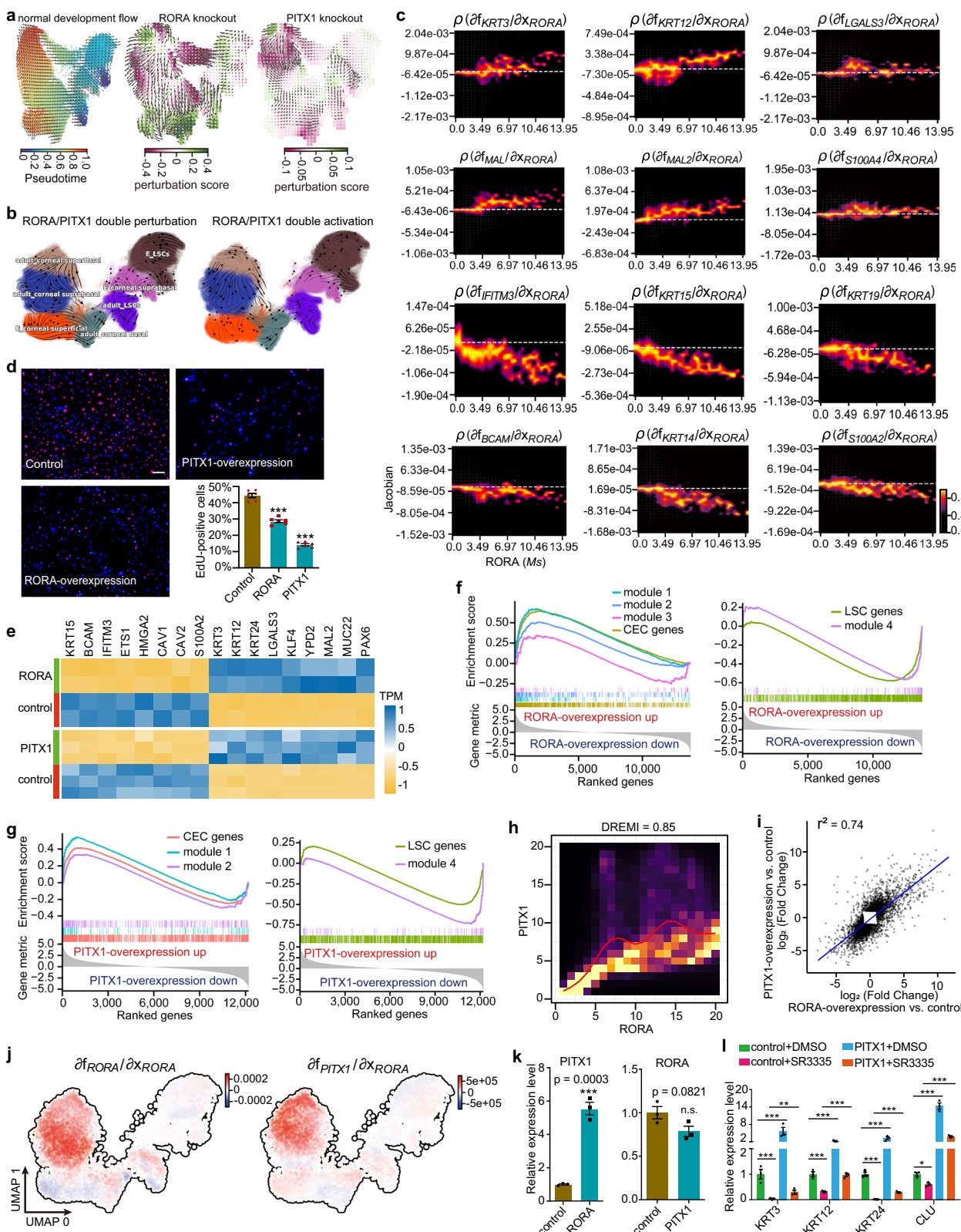

and *PITX1* exhibited bivalent state and were further silenced by distal H3K27me3 sites that looped to their promoters (Fig. 4h). These H3K27me3-mediated interactions were disrupted when RORA was ectopically expressed. In addition, RORA activated itself by decommissioning H3K27me3 at the promoter and simultaneously connecting the promoter to a nearby H3K27ac/H3K4me3-enriched active promoter bound by RORA (Fig. 4h). Following binding to the promoter of

*PITX1*, RORA induced the conversion of H3K27me3 to H3K27ac and further established a new P–P interaction (Fig. 4h), which led to *PITX1* activation. To activate *KRT12* and *KLF4*, RORA bound and increased H3K27ac levels at their promoters, disrupted the promoter-H3K27me3 loops, and anchored their promoters to pre-established distal enhancers (Fig. 4i). RORA up-regulated the expression of *LGALS3* by establishing a new interaction between the promoter and enhancer that was

**Fig. 3 | RORA-PITX1 axis initiates and drives LSCs to differentiate into mature CECs. a** CellOracle *RORA* and *PITX1* knockout simulation vectors with perturbation scores and normal development vector. **b** Dynamo in silico double perturbation and double activation simulation vectors of *RORA* and *PITX1*. **c** Dynamo response heatmaps showing Jacobian regulation from *RORA* to the selected genes versus *RORA Ms*. White dashed lines indicate the zero value. Positive value represents activation from *RORA* to the indicated genes and negative represents repression from *RORA* to the indicated genes. **d** 5-Ethynyl-20-deoxyuridine (EdU) staining for LSCs treated with control, *RORA*- and *PITX1*-overexpression. Bar plots showing statistics on the percentage of EdU-positive cells for each group. Scale bar, 100 µm. Data are represented as means ± SEM (*n* = 6 independent experiments, ****P* < 0.001). SEM: Standard Error of the Mean. *P*-values were calculated using two-sided unpaired Student's *t* tests. Source data are provided as a Source Data file. **e** Heatmap of bulk RNA-seq data showing the selected differentially expressed genes upon *RORA* and *PITX1* overexpression in LSCs. **f, g** GSEA (Gene Set Enrichment Analysis) for the indicated gene sets using bulk RNA-seq data of *RORA*- and *PITX1*-overexpression versus control. **h** Single cell kNN-DREMI analysis (DREVI plots and DREMI values) indicates the relationship between *RORA* and *PITX1* after imputation with MAGIC. **i** Linear regression test for gene fold changes induced by *RORA* and *PITX1* overexpression. **j** Dynamo Jacobian from *RORA* to *RORA* and *PITX1* in the UMAP space. Positive value (red) indicates positive regulation. **k** Real-time quantitative PCR (qRT-PCR) analysis for *RORA* upon *PITX1* overexpression and *PITX1* upon *RORA* overexpression. Data are represented as means ± SEM (*n* = 3 independent experiments, ****P* < 0.001). n.s., nonsignificant. *P*-values were calculated using two-sided unpaired Student's *t* tests. Source data are provided as a Source Data file. **l** qRT-PCR analysis for *KRT3, KRT12, KRT24* and *CLU* in CECs treated with the indicated groups for one week upon differentiation from LSCs. Data are represented as means ± SEM (*n* = 3 independent experiments, **P* < 0.05, ***P* < 0.01, ****P* < 0.001). *P*-values were calculated using two-sided unpaired Student's *t* tests. Source data are provided as a Source Data file.

switched from the H3K27me3 mark (Fig. 4i). Conversely, RORA silenced the promoter of LSC-specific *EMP3* and disrupted the *EMP3*-anchored P–P loop, leading to a decrease in the expression of *EMP3* (Fig. 4i).

In summary, RORA orchestrates the CEC differentiation program by activating lineage-specific promoters and/or enhancers, disrupting H3K27me3-anchored chromatin loops, and connecting promoters to distal active regulatory elements (Fig. 5). RORA inhibits LSC genes by silencing their promoters and/or disrupting promoter-anchored chromatin interactions (Fig. 5).

## Discussion

The corneal epithelium is a stratified squamous epithelial tissue that plays a crucial role in ocular health[2]. It is continuously renewed with a typical turnover time of 5–7 days and heals rapidly after an injury, which are dependent on a population of LSCs located at the basal layer of the limbus[1,5]. The gene regulatory network underlying the differentiation process from LSCs to CECs has not been well-documented yet. Recent advances in single-cell transcriptomics provide an unprecedented avenue to systematically appreciate genetic and functional heterogeneity of complex biological systems[34–36]. In this study, we separated and annotated each cell type in the limbal–corneal epithelium based on a single-cell atlas of the developing cornea (Fig. 5). Identification of gene co-expression modules and SCENIC TF activities underscored cell type- and stage-specific gene regulatory circuitries driving CEC development and differentiation (Fig. 5).

LSCs can differentiate into basal, suprabasal and superficial cells of the corneal epithelium[5]. Single-cell profiles allow for inferring cell-state transition and molecular dynamics during tissue development and differentiation[37–39]. In this study, RNA velocity and pseudotime estimations showed that the degree of cell differentiation progressively increased in the order of LSC, basal cells, suprabasal cells and superficial cells. We also revealed the gene acceleration kinetics that was consistent with gene function during CEC differentiation progression. This study provided detailed insights into the molecular dynamics of corneal epithelial development and differentiation, which facilitates us to better understand corneal homeostasis and pathogenesis.

It has been well-established that TFs dictate stem cell self-renewal and differentiation[40]. Thus, there is an urgent need to identify the master regulators of CEC differentiation. CellRank analysis of our scRNA-seq data indicated a set of TFs as potential CEC differentiation drivers, including RORA, PITX1, KLF4, KLF5 and EHF. We found that RORA and PITX1 were specifically expressed in the suprabasal and superficial cell layers. Both in silico gene perturbation simulations and in vitro experimental verification showed that RORA could turn off LSC-specific genes and initiate CEC differentiation program via activating PITX1. Therefore, we identified the RORA–PITX1 axis as the

molecular switch that dictated CEC differentiation. Interestingly, a previous report suggested that RORA strongly inhibits the induction of induced pluripotent stem cells from human CECs[41], which further indicates that RORA plays a key role in maintaining CEC differentiation state.

In addition, in silico gene perturbation simulation using Dynamo or CellOracle with the learned vector in our single-cell data enables the large-scale prediction of gene function in human limbal–corneal epithelial tissue. It is well-known that cell cycle withdraw is required for stem cell differentiation. Interestingly, RORA and PITX1 also downregulated the proliferation-related genes like *ETS1* and *HMGA2*, resulting in reduced LSC proliferation. Thus, the inhibition of LSC proliferation induced by RORA and PITX1 is important for LSC differentiation. Our previous study revealed that RUNX1, PAX6 and SMAD3 are expressed in both LSCs and CECs and play an indispensable role in corneal epithelial homeostasis[42]. Loss of them not only disrupted CEC differentiation, but also induced cellular fate transition by activating the epidermal genes[42]. In contrast, depletion of *RORA* and *PITX1* only blocked CEC differentiation, but did not induce cellular fate switch. Thus, RUNX1, PAX6 and SMAD3 are required for corneal epithelial fate determination, while RORA–PITX1 axis is important to corneal epithelial differentiation.

Chromatin three-dimensional interactions and epigenetic architecture that are shaped by lineage-specific TFs underlie cellular fate and identity[43–45]. Our previous publication dissected the genome-wide chromatin three-dimensional organization and epigenetic architecture in LSCs using Hi-C sequencing[46]. We showed that the LSC fate determinants RUNX1, PAX6 and SMAD3 govern LSC identity through coupling of long-range chromatin interactions and chromatin epigenome[42,46]. We also revealed the TP63-mediated chromatin loops that were involved in LSC self-renewal[46]. In this work, we showed that RORA altered histone modifications of the cis-regulatory elements, which shaped CEC-specific enhancers and promoters. Using promoter capture Hi-C approach, we found that RORA activated CEC genes by establishing CEC-specific P–P and enhancer-promoter loops. RORA turned off LSC genes by silencing their promoters and/or disrupting promoter-anchored interactions. Therefore, our data highlighted a RORA-mediated three-dimensional epigenetic regulatory circuit underlying corneal epithelial differentiation (Fig. 5). The dissection of molecular mechanism underlying LSC self-renewal and differentiation provided comprehensive insights into corneal epithelial biology.

## Methods

### Human limbus samples

All normal human limbus tissues were obtained as de-identified surgical specimens from eyebank of Zhongshan Ophthalmic Center. This study was conducted in accordance to the criteria set by the

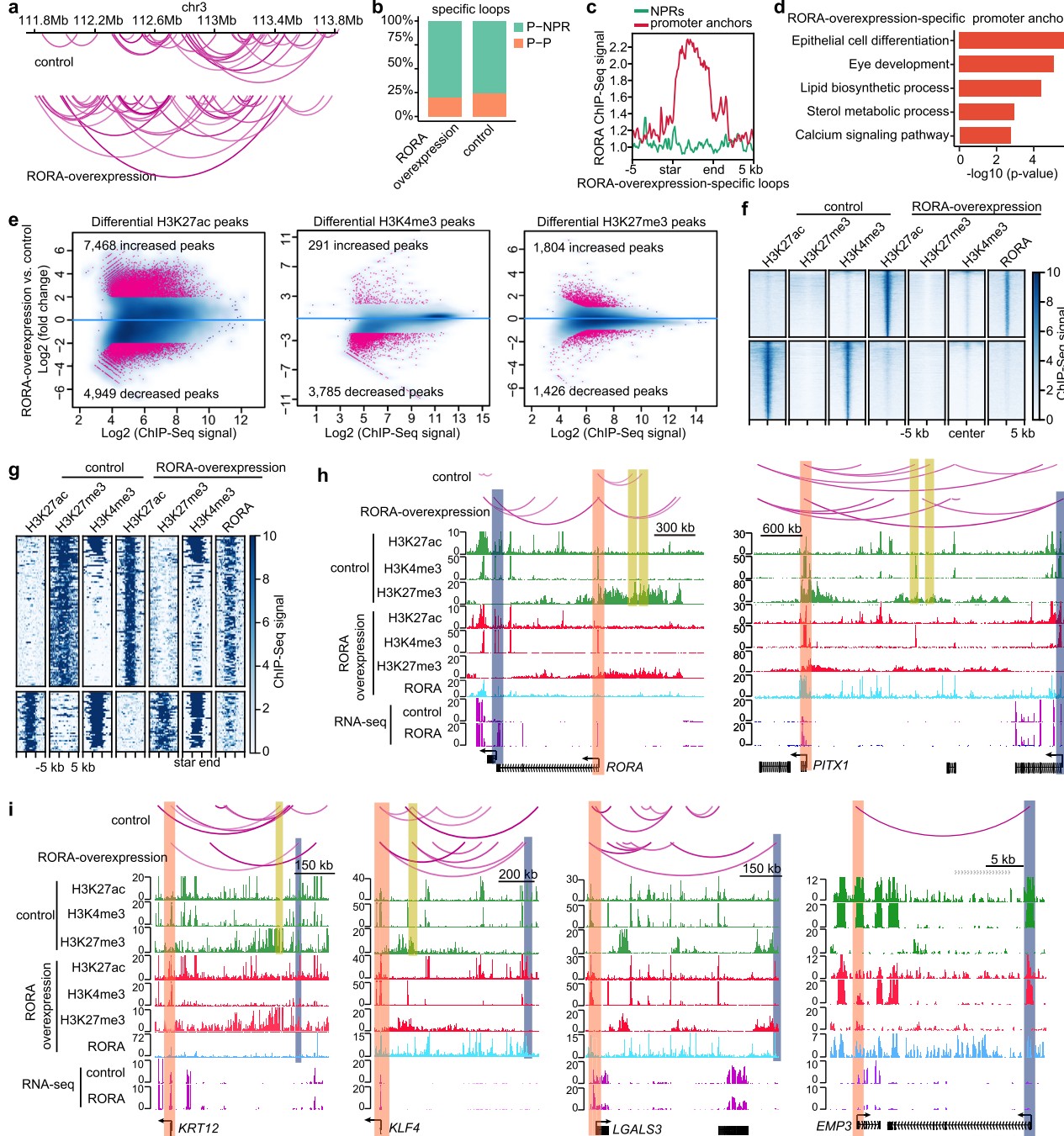

**Fig. 4 | RORA-mediated epigenetic remodeling underlies CEC differentiation.**
**a** Selected tracks for promoter-anchored chromatin interactions in control and *RORA*-overexpressed LSCs. **b** Bar plots showing the percentages of P-P and P-NPR loops respectively in *RORA*-overexpression- and control-specific interactions. **c** RORA ChIP-seq signal at promoter and NPR anchors of *RORA*-overexpression-specific interactions. **d** GO terms enriched in *RORA*-overexpression-specific promoter anchors. GO enrichment analysis were performed and *P*-values were calculated in Metascape[52]. **e** Scatterplots showing the differential H3K27ac, H3K4me3 and H3K27me3 peaks upon *RORA* overexpression. **f** Heatmaps showing the indicated ChIP-seq signals across the peaks with H3K27ac reduction and increase upon *RORA* overexpression. **g** Heatmaps showing the indicated ChIP-seq signals across the peaks with H3K27ac and H3K27me3 shifts upon *RORA* overexpression. **h, i** Genome browser tracks for promoter-anchored chromatin interactions and the indicated ChIP-seq and RNA-seq signals across the *RORA, PITX1, KRT12, KLF4, LGALS3* and *EMP3* loci in control and *RORA*-overexpressed LSCs.

Declaration of Helsinki and was approved by the Ethics Committee of Zhongshan Ophthalmic Center of Sun Yat-sen University.

## Human primary LSC culture and in vitro differentiation

After digestion with 0.2% collagenase IV (Gibco, # 17104019) at 37 °C for 2 h and 0.25% trypsin-EDTA (Gibco, #25200056) for another 15 min, the donors' limbal ring biopsies were cut into small pieces and cultured on polystyrene plates (Corning) precoated with Matrigel (BD Biosciences, #354230). The components of LSC medium[15,42] included DMEM/F12 and DMEM (1:1) with 1% penicillin/streptomycin (Gibco, #15140122), 10% fetal bovine serum (Gibco, #16000-044), 10 ng/ml EGF (millipore, #GF144), 5 μg/ml insulin (sigma, #I5500), 0.4 μg/ml

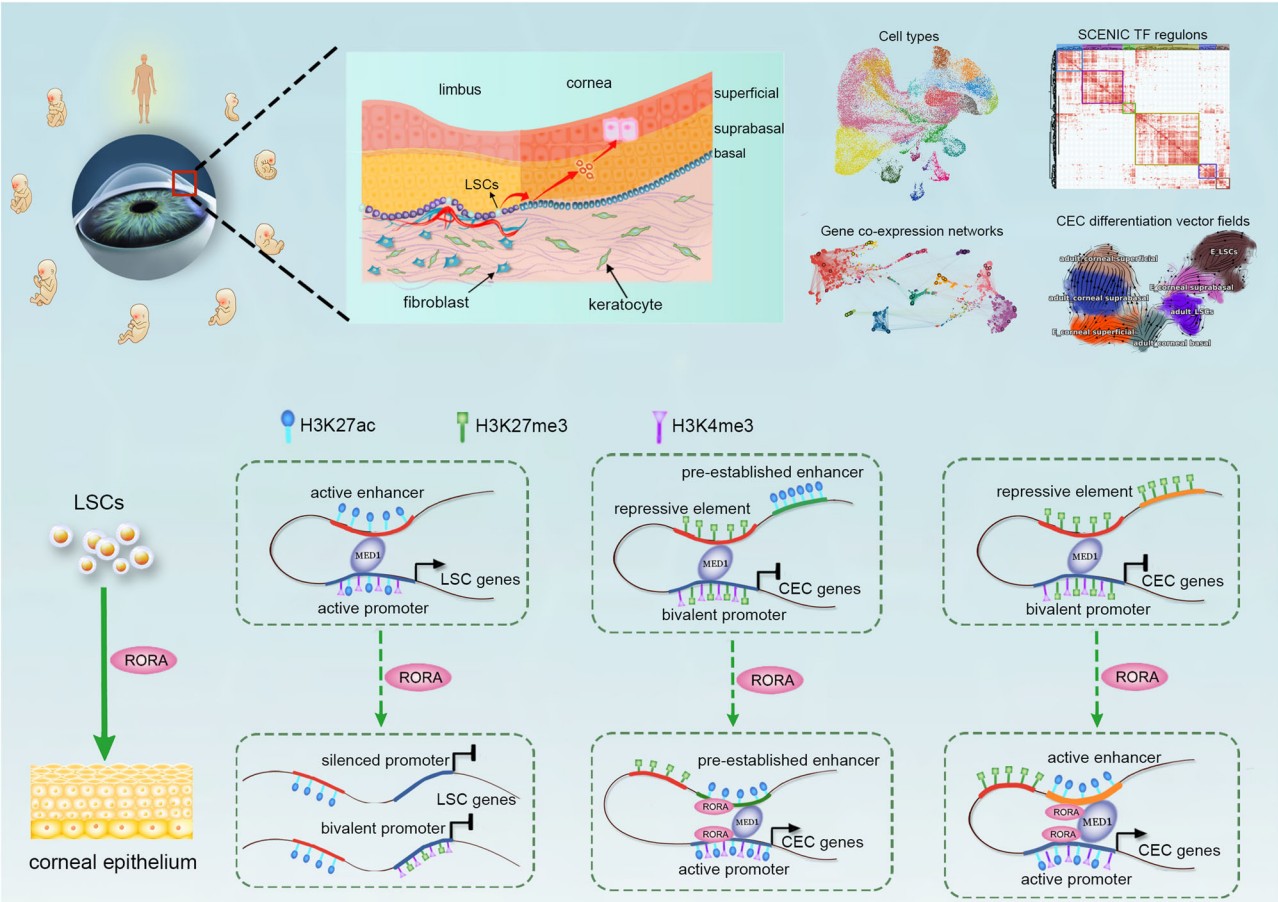

**Fig. 5 | Schematic overview of the study.** We delineate single-cell transcriptomic dynamics underlying corneal epithelial differentiation. RORA drives the expression of CEC genes by activating lineage-specific promoters and/or enhancers, disrupting promoter-H3K27me3 interactions, and established new P–P or enhancer-promoter interactions. RORA inhibits LSC genes by silencing their promoters and/or disrupting promoter-anchored chromatin interactions.

hydrocortisone (Millipore, #386698), 0.1 nM cholera toxin (sigma, #C8052), and 2 nM 3,3′,5-triiodo-L-thyronine (sigma, #T2877). The medium was replaced with a fresh one every day. In order to induce in vitro differentiation of LSCs, after LSC confluence, the medium was switched to the defined K-SFM medium (Gibco, #10744-019) with 1.2 μM calcium for one week.

## Gene knockdown and overexpression
shRNAs specific for PITX1 (5′-TCAACGCGTGCCAGTACAACA-3′, 5′-CTCGGGCCTCAACAACATCAA-3′) and scrambled shRNAs (5′-CCTAAGGTTAAGTCGCCCTCG-3′) were cloned to PLKO.1 plasmids. cDNAs encoding flag-tagged RORA and PITX1 proteins were cloned into pCDH plasmids and the empty vector was used as control. These plasmids were transfected into HEK293T cells to produce lentivirus for three days using Lipofectamine 3000 kit (Invitrogen, #L3000015) according to the manufacturer's instructions. Cells were infected with lentiviral particles for 24 h and selected with 2 μg/mL puromycin (Invitrogen, #A11138-03) for 48 h.

## RNA extraction and qRT-PCR
Total RNA was isolated from cells using the RNeasy Mini kit (Qiagen, #74106) and reverse transcription polymerase chain reaction was performed to synthesize cDNA using the PrimeScrip RT Master Mix Kit (Takara, #RR036A). qRT-PCR was conducted using the iTaq Universal SYBR Green Supermix Kit (Bio–Rad, #1725125). Relative gene expression level was normalized to that of GAPDH.

## Immunofluorescence staining
Immunofluorescence staining experiments were performed as previously described[42]. In brief, tissue samples were successively fixed, decalcified, embedded in paraffin and deparaffinized, followed by antigen repair. Cells were fixed in 4% paraformaldehyde directly. The tissue sections or cells were permeabilized and blocked with 0.3% Triton X-100 and 3% BSA in PBS for 1 h, followed by incubation with primary antibodies overnight at 4 °C and secondary antibodies for 1 h at room temperature. All images were obtained using ZEISS LSM 800 confocal laser scanning microscope.

The antibodies used for immunofluorescence are as follows: anti-KRT3 (Abcam, #ab68260, 1:200), anti-KRT12 (Abcam, #ab124975, 1:200), anti-LGALS3 (Biolegend, #125401, 1:200), anti-S100A2 (Abcam, #ab109494, 1:200), anti-KRT15 (NeoMarkers, #MS-1068-P0, 1:200), anti-IFTIM3 (Proteintech, #11714-1-AP, 1:200), anti-PAX6 (Sigma, #AMAB91372, 1:200), anti-ALDH3A1 (GeneTex, #GTX30042, 1:200), anti-CLU (Proteintech, #12289-1-AP, 1:200), anti-KRT19 (Biolegend, #628502, 1:200), anti-RORA (Immunoway, #YT4166, 1:100), anti-PITX1 (Sigma, #HPA008743, 1:100), Anti-KRT14 (Thermo, #MA511599, 1:200), Anti-KI67 (Cell Signaling Technology, #9129 S, 1:200), Anti-TP63 (Cell Signaling Technology, 67825 S, 1:200), anti-rabbit IgG (Alexa Fluor 488 Conjugate, CST, 4412 S,1:1000), anti-mouse IgG (Alexa Fluor 488 Conjugate, CST, 4408 S,1:1000), anti-rabbit IgG (Alexa Fluor 594 Conjugate, CST, 8889 S,1:1000), and anti-mouse IgG (Alexa Fluor 594 Conjugate, CST, 8890 S,1:1000).

## EdU assay

For cell proliferation assay, the BeyoClick EdU Cell Proliferation Kit with Alexa Fluor 555 (Beyotime, #C0075S) was used. Cells were seeded in 6-well plate and treated with EdU reagent for 2 h. After washing in PBS, cells were fixed, permeated, and incubated with Click Additive Solution and DAPI successively following the manufacturer's instructions.

## ChIP-Seq

ChIP-Seq were performed as previously described[42]. Cells were fixed in 1% formaldehyde (Invitrogen, #28906) for 10 min at room temperature and quenched with 0.125 M glycine for 5 min. The cross-linked cells were sonicated in lysis buffer (50 mM HEPES−NaOH, pH 7.5, 500 mM NaCl, 1 mM EDTA, 0.1% Na-deoxycholate, 1% TritonX-100 and 0.1% SDS) to shear the chromatin DNA into 200- to 400-bp lengths. Chromatin was immunoprecipitated with primary antibodies overnight at 4 °C in IP buffer (HEPES−NaOH, pH 7.5, 300 mM NaCl, 1 mM EDTA, 0.1% Na-deoxycholate, 1% TritonX-100 and 0.1% SDS). Protein A and G Dyna-beads (1:1, Invitrogen,) were added to extract the immunoprecipitated chromatin complex and were washed sequentially with lysis buffer, low-salt buffer (10 mM Tris−HCl, pH 8.0, 250 mM LiCl, 1 mM EDTA, 0.5% NP40 and 0.5% Na-deoxycholate) and TE buffer (10 mM Tris−HCl, pH 8.0, and 1 mM EDTA). Immunoprecipitated complex was eluted from beads in elution buffer (50 mM Tris-HCl pH 8.0, 10 mM EDTA and 1% SDS) at 65 °C for 4 h and then digested with 20 µg RNase A (Invitrogen, #EN0531) and 40 µg Proteinase K (Invitrogen, #AM2546) at 55 °C for 1 h. DNA was purified using the MinElute PCR Purification Kit (Qiagen). ChIP-seq DNA libraries were generated using VAHTS Universal DNA Library Prep Kit for Illumina V3 (Vazyme) according to the manufacturer's instructions and sequenced using the NovaSeq 6000 Platform. The antibodies used for ChIP-Seq (5 µg/ChIP) are as follows: anti-H3K27ac (Millipore, #07-360), anti-H3K27me3 (Cell Signaling Technology, #9733), anti-H3K4me3 (Cell Signaling Technology, #9751), anti-FLAG (Cell Signaling Technology, #14793).

## Promoter capture Hi-C

Cells were cross-linked with 1% formaldehyde at room temperature for 10 min and quenched with 0.2 M glycine for 5 min. After cellular lysis, 0.3% SDS was added to deactivate endogenous nucleases. Then, the chromatin DNA was digested with 100 U HindIII (NEB, #R3104S) and labeled with biotin-14-dCTP (Invitrogen, #19518018). The biotin-labeled DNA fragments were then ligated by 50 U T4 DNA ligase (NEB, # M0202S), de-crosslinked and extracted using QIAamp DNA Mini Kit (Qiagen). We then sheared the purified DNA fragments to 300−600 bp, repaired the ends and added A-tails and SureSelect adaptors. After biotin-streptavidin-mediated pull-down, DNA fragments were amplified by PCR. We conducted capture Hi-C of promoters by the SureSelect XT Library Prep Kit ILM (Agilent Technologies, #G9981A), as per the manufacturer's instructions, with custom-designed biotinylated RNA bait library and custom paired-end blockers. After another PCR amplification, promoter capture Hi-C libraries were purified with AMPure XP beads (Beckman Coulter, #a63880) and sequenced on the NovaSeq 6000 platform.

## Bulk RNA-seq data analysis

Bulk RNA-seq data were analyzed as previously described[15,42]. Briefly, the sequencing reads were trimmed using the trimmomatic tool[47] and then mapped to the human hg19 reference genome using the STAR package[48]. The transcripts per kilobase million (TPM) values were calculated using RSEM algorithm[49]. DESeq2 R package[50] was utilized to identify the differentially expressed genes with fold change ≥ 2 and *p*-value < 0.05. GSEA[51] of the bulk RNA-seq data was performed in R, with a *p*-value cut-off of 0.05. GO enrichment analysis were performed in Metascape[52].

## ChIP-seq data analysis

ChIP-seq data were analyzed as previously described[42]. Briefly, BWA algorithm[53] was used to map the trimmed sequencing reads to the human hg19 genome, and duplicated reads were removed using Picard MarkDuplicates (http://broadinstitute.github.io/picard/). Peak calling was performed using MACS2[54] with a *q*-value cut-off of 0.01. MACS2 parameters were set as follows: for H3K27ac and H3K4me3 data, "−extsize 200" was set; For H3K27me3 data, "−extsize 500, −broad" was set. Heatmaps of ChIP-seq signals were presented through deepTools[55]. ChIP-seq signal tracks were visualized through Integrative Genomics Viewer software[56]. GSEA of H3K27ac ChIP-seq data was performed using the GSEA Java software. We employed the DiffBind[57] R package to identify differential peaks with a log fold change ≥ 2 and FDR < 0.01.

## Promoter capture Hi-C data analysis

Promoter capture Hi-C data analysis was performed as previously described[58]. Briefly, HiCUP[59] pipeline was used for capture Hi-C data processing, including read mapping, read filtering and removal of duplicate reads. CHiCAGO[60] was used to identify significant chromatin interactions (CHiCAGO score ≥ 5). Significant differential interactions between the two groups were determined using edgeR with |log fold change | > 1 and *p*-value < 0.05. The WashU EpiGenome browser[61] was used to visualize the chromatin interactions.

## scRNA-seq data processing and visualization

The raw scRNA-seq data were obtained from the Gene Expression Omnibus under the accession number GSE155683[16]. Sequencing reads were mapped onto the human GRCh38 genome using Cell-Ranger software. Count matrices of the unique molecular identifiers were processed using Seurat R package[62]. Cells with more than 15% mitochondrial reads and <300 genes were excluded. The *NormalizeData* function in the Seurat package was used to generate log-normalized expression values for the raw counts. For the ocular surface data, *SCTransform* function was used to normalize and scale the expression matrices and identify highly variable genes in each sample. Following canonical correlation analysis-based integration for all datasets using *FindIntegrationAnchors* (k.anchor = 10) and *IntegrateData* functions, we performed principal component analysis (PCA) dimensionality reduction and graph-based cell clustering using the *FindNeighbors* (dims = 20) and *FindClusters* (resolution = 1.5) functions. The *RunUMAP* function was used to generate UMAP maps using the top 20 principal components. Signature genes for each cell population were identified by *FindAllMarkers* function (min.pct = 0.5, logfc.threshold = 0.5 and test.use = "wilcox") and retained with *p*-value < 0.05. To further clustering the limbal and corneal epithelial compartments, the scanpy toolkit[63] was used to perform PCA, compute neighborhood graphs, integrate data using BBKNN and cluster the neighborhood graphs with the default parameters. Latent time analysis and terminal differentiation-associated TFs were inferred through CellRank[20], according to the official tutorials with default parameters (https://cellrank.readthedocs.io/en/stable/).

## hdWGCNA analysis

The hdWGCNA package[24] was used to perform gene co-expression network analysis of the limbal−corneal epithelial cell types. After setting up Seurat object for WGCNA, metacells were constructed with the following parameters: k = 25, assay = 'SCT', max_shared = 20, min_cells = 300, reduction = 'umap', slot = 'data'. Only cells within the same cell type in the same sample were pooled for metacell construction. *ConstructNetwork* function was used to construct co-expression network with a soft-power threshold of 8. GO enrichment analysis of each module was performed using EnrichR package.

## SCENIC analysis

pySCENIC[26] was used to identify cell-type-specific gene regulatory networks in the limbal–corneal epithelial scRNA-seq dataset. The raw UMI count expression matrix in Seurat was used as input data. We filtered out genes expressed in > 50 cells. The genes available in the RcisTarget human hg38 motif databases were used. We then constructed a co-expression network by grnboost2, inferred regulons for each TF using the RcisTarget motif databases (hg38-refseq-r80-10kb-up-and-down-tss.mc9nr.genes-vs-motifs.rankings.feather and hg38-refseq-r80-500bp-up-and-100bp-down-tss.mc9nr.genes-vs-motifs.-rankings.feather), and calculated activity score for each TF regulon in each cell by AUCell.

## Dynamo analysis

The spliced and unspliced expression matrices were generated using the Velocyto Python package[64]. The total, spliced and unspliced RNA expression matrices were stored as an AnnData file, which was then used in Dynamo[28]. After normalizing all datasets in different layers and performing PCA dimensionality reduction using the *recipe_monocle* function, we applied *dyn.tl.dynamics* to estimate the expression dynamics and learn the velocity for each gene. The estimation of ddhodge potential, cell acceleration, cell divergence, cell-state transition, RNA Jacobian and in silico perturbation was conducted according to the official tutorials with default parameters (https://dynamo-release.readthedocs.io/en/latest/index.html).

## CellOracle simulation

We used CellOracle[31] to simulate cell identity shifts following TF knockout and to calculate the pseudotime and perturbation scores based on a SCENIC TF regulatory network constructed in our limbal-corneal epithelial scRNA-seq data. The CellOracle analysis was conducted according to the official tutorial with default parameters (https://morris-lab.github.io/CellOracle.documentation/).

## Statistics and reproducibility

Statistical analysis was conducted using the Student's unpaired two-tailed t-test (for comparison between two groups) and ANOVA (for multiple comparisons) with GraphPad Prism (v7.0). The results are presented as the mean ± SEM (*$P < 0.05$; **$P < 0.01$; ***$P < 0.001$; n.s., nonsignificant). The sample sizes ($n$) are indicated in the figure legends. No statistical method was used to predetermine sample size. No data were excluded from the analyses.

## Reporting summary

Further information on research design is available in the Nature Portfolio Reporting Summary linked to this article.

## Data availability

The scRNA-seq data used in this paper were downloaded from the Gene Expression Omnibus under the accession number GSE155683. The raw sequence data generated in this paper have been deposited in the Gene Expression Omnibus under the accession number GSE249150. The RcisTarget motif databases are available at https://resources.aertslab.org/cistarget/databases/homo_sapiens/hg38/refseq_r80/mc9nr/gene_based/. Source data are provided with this paper.

## Code availability

The code of bioinformatics analysis in this paper is available through GitHub (https://github.com/Mingsenli/corneal-single-cells, https://doi.org/10.5281/zenodo.10211882).

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

## Acknowledgements

This work was supported by National Natural Science Foundation of China (NO. 32270782 to M.L. and NO. 82271043 to H.O.), National Natural Youth Science Foundation of China (NO. 32100449 to M.L.), Natural Science Foundation of Guangdong Province (No. 2022A1515012622 to M.L.), and Projects of International Cooperation and Exchanges NSFC (No. 32061160364 to H.O.). We thank Wuhan Frasergen Biotechnology Co., Ltd for assisting in promoter capture Hi-C experiment and bioinformatics analysis.

## Author contributions

H.O., M.L. and J.J. designed and guided this project. M.L. performed ChIP-Seq experiments, bioinformatics analysis and wrote the manuscript. H.G., B.W., Z.H. and S.W. performed cellular and molecular biology experiments. H.H., F.A., Z.L., K.M., J.Z. and J.L. performed cell culture. J.T., C.L., L.W., X.D. and G.L. analyzed experiments and supervised the manuscript.

## Competing interests

The authors declare no competing interests.
