## [Peer Review File · Nature Communications]

The single-cell transcriptomic atlas and RORA-mediated 3D epigenomic remodeling in driving corneal epithelial differentiationREVIEWER COMMENTS

Reviewer #1 (Remarks to the Author):

This study investigated the transcription dynamics of human limbal stem cells (LSCs) differentiation into corneal epithelial cells (CECs). By using the publicly available single cell RNA sequencing datasets, the authors successfully delineated a mechanism of RORA and PITX1 mediated histone post-translational modification, which further regulated the 3D chromosomal interactions to control the expression of CEC genes. This manuscript is well-written with advanced data analyses, supported by some validation cellular experiments. Here are my major comments:

1. Line 68-69: Please also provide some examples about what other cell types also express these markers (KLF4, PAX6, EHF and FOXC1).
2. Line 106: There are only 10 cell populations in Fig 1A, instead of 11 cell populations as stated in the main texts.
3. Please indicate the locations of limbus and central cornea in Fig 1D.
4. In Fig 1G, please also show the p-values of these GO terms.
5. Line 136: About the statement of “canonical markers specific to the differentiating adult corneal epithelium and was involved in the response to cadmium, copper and zinc ions”, please cite some references to support this claim.
6. Line 141: Why only focus on these 3 hub genes but not others like SPARC, GPC3, CRABP2, RPS3, RPL6, EEF1A1 and MEG3?
7. Line 186: There are 21 TFs in Fig 2A, instead of 20 TFs as stated in the main texts.
8. Line 193: Please indicate RORA and PITX1 in Fig 2C.
9. Again please indicate the locations of limbus and central cornea in Fig 2F.
10. Line 261: It is very interesting to see proliferation, in addition to differentiation, was also affected in PITX1-OE and RORA-OE. The authors could discuss why cell proliferation is also affected.
- 11: One minor comment on line 417: The word “underly” should be written as “underlie”.

Reviewer #2 (Remarks to the Author):

The manuscript presents an integrative study that combines a reanalysis of previously published single-cell data with new molecular biological techniques to explore differentiation in the corneal epithelium. This multi-faceted approach adds depth and breadth to the investigation and strengthens the study.

However, there are some concerns in the manuscript (see specific points below):

Please discuss the relationship with previous reports in light of the current single-cell results, which have indicated that RUNX1, PAX6, and SMAD3 cooperatively interact with each other and establish core transcriptional regulatory circuits in limbal stem/progenitor cells.

Previous report revealed that RORA strongly inhibited iPS induction from human corneal epithelial cells, indicating that RORA has a potential role in maintaining differentiated corneal epithelial cells (Cell Rep. 2016 May 10;15(6):1359-68. (PMID: 27134177)). It would be engaging for the authors to discuss it.

It would be interesting to understand the cellular fate by showing what happens to cell proliferation-related or cell-cycle-related genes as the cells move toward the surface.

Introduction.

Authors should add the key references regarding transcription networks of corneal epithelial cells and limbal epithelial stem cells. (PAX6: Exp Eye Res. 2017 Jan;154:30-38 (PMID# 27818314), ABCB5: Nature. 2014 Jul 17;511(7509):353-7 (PMID: 25030174))

Line193-195: The chart does not indicate each gene's Log2FC values. This makes it difficult to assess the magnitude and significance of gene expression changes.

Line 646: While the t-test is commonly used for comparing two groups, it is not appropriate for multiple comparisons, as is the case in the current study.

Figure 1C: It shows no difference in the expression of LSGALS3 in the PCW18 cell group compared to the other groups at other times a few weeks earlier than PCW18. Could the authors clarify the basis for their assertion that LSGALS3 is expressed in this group?

Figure 2C: The volcano plots aim to present the differential gene expression data. However, it is unclear which plots correspond to which genes.

Figure 2J, 3H: The manuscript utilizes the kNN-DREMI score as both the vertical and horizontal axes. While the kNN-DREMI score is recognized for measuring relationships between genes, its use in this manner might not be appropriate.

Figure 3E: The label on the vertical axis of the figure is marked as "ROR" when it appears that it should be labeled "RORA" to reflect the gene accurately.

Figure 3L: The authors claim that OE significantly rescues the expression suppression caused by SR3335, but statistical evidence does not support this assertion.

Fig S4: If the target sequence for each transcription factor or enhancer region is known, please add it to the chart.

Fig 4H, 4I: Please add a description of the vertical scale. Chip-seq detects peaks, but authors should consider whether the positions of the peaks are consistent with the actual binding sequences of TFs.

Figure 5: In the upper right corner of Figure 5, there appears to be a plot that suggests the representation of TADs. However, this part of the figure is not discussed in either the figure legend or the main text.

Reviewer #3 (Remarks to the Author):

In the report, the authors performed analysis of previously published single cell RNA-seq data for development and adult cornea and identified RORA and PITX1 as key factors for promoting corneal epithelia cell (CEC) development from limbal stem cells (LCS). To test this prediction, over expression and knock down of RORA and PITX1 in primary LSC culture have been tested. Furthermore, CHIP-seq of RORA and PITX1 along with histone epigenetic marks and HiC experiments are used to investigate the mechanisms of gene regulation. The author's effort on experimentally testing the hypothesis is great and the results are very

encouraging. The overall results are largely consistent with the informatic analysis from single cell data and the connection of RORA and PITX1 in CEC development is interesting. My specific comments are the following:

1. The authors reanalyzed previously published datasets. In figure 1A's UMAP, cells from PCW17 appear to segregate, forming the 'E_corneal superficial' cluster. This could be indicative of an uncorrected batch effect. Indeed, the RNA velocity analysis in figures 1I and 1K portrays potentially inconsistent trajectory to this cluster. Exploring alternative meta-analysis tools like scVI or Harmony might yield more coherent clustering.
2. RORA and PITX1 don't exhibit high expression until later developmental stages and are predominantly expressed in terminally differentiated CECs. This would lead one to anticipate that these factors play roles in CEC terminal differentiation rather than as precursory factors transitioning LSC to CEC. How do the authors reconcile this seeming discrepancy? Is there any evidence of these genes being active in proliferating progenitor cells derived from LSC in the adult cornea?
3. While the overexpression of RORA and PITX1 clearly induces marker gene expression characteristic of CECs and decrease in LSC proliferation in culture. Given their roles as transcription factors, it's not unexpected for their overexpression to instigate a cascade of gene expression. Yet, it's crucial to establish if their overexpression truly facilitates the LSC to CEC developmental transition. Validating that cultured cells undergo accelerated CEC development and result in genuine CECs is essential.
4. Given the inherent differences between in vitro and in vivo settings, corroborating the central hypothesis using animal models like mice or zebrafish, especially through gene knockouts or knockdowns during development, would be critical.

Reviewer #4 (Remarks to the Author):

This is an interesting study where the authors use a single-cell transcriptomic atlas to show the molecular dynamics during human corneal epithelial cell development and differentiation. They show the role of RORA and PITX1, primarily in silico in driving limbal stem/progenitor cells to differentiate into corneal epithelial cells. Overall the study is of interest and well-performed. Comments are below.

- 1) At the low level of resolution and image quality, it is challenging to determine co-localization of the limbal stem cell maker S100A2 and KRT15. The authors need to provide higher quality images at greater magnification. It is also necessary to perform quantification showing the degree of overlap between S100A2 and KRT15.
- 2) The authors claim that limbal stem cells express IFITM3; however, in the right panel there is no co-localization with a marker. The authors need to perform double immunofluorescence with a limbal stem cell marker to make this claim.
- 3) The authors should refrain from using short-hand terms. For example, the text states that the limbal stem cell markers KRT15 and KRT14 were used (line 115), whereas the figure states K12 and K15. It's confusing why the text refers to KRT14; however, the figure states K12? The authors should clarify this. There is no notation in the figure legend or text that K12 or K15 refer to KRT15 and KRT14. Thus the authors need to write out all gene names in full. The authors should also write out corneal epithelial cells and limbal stem/progenitor cells each time rather than CECs for people outside the field who are not familiar with these terms. Similarly they should write out overexpression rather than the term OE.
- 4) In the text the authors state that LSGALS3 was primarily expressed in the suprabasal and superficial layers of the limbus and cornea; however, in the figure uses the term LGALS3. I assume the authors are referring to LGALS3, and this appears to be a mistake in the text,

which should be corrected. The authors need to provide a higher resolution image in Figure 1D adult for LGALS3, as it's not possible to make out the expression in the suprabasal and superficial layers at low resolution.

5) The authors need to annotate all the figures of the corneas to show where the suprabasal, superficial, and limbal-corneal regions are. This is important to perform as we cannot tell the regions or corneal histology based on DAPI nuclear staining alone.

6) The authors state that PITX1 is expressed in the differentiated limbal-corneal epithelial layer of adults; however, no co-staining is performed. This should be performed with markers. Similarly we cannot determine if there is co-localization of RORA and KRT3 at the low level magnification. Quantification should be performed on co-localization studies.

7) The authors make the claim "In a word, we highlighted that RORA-PITX1 axis is necessary and sufficient to dictate corneal epithelial differentiation." Most of the studies are performed in silico with the exception of in vitro studies of limbal stem cells. The authors need to perform conditional knockout in mouse of RORA-PITX1 to make the claim that the RORA-PITX1 axis is necessary and sufficient to dictate corneal epithelial differentiation. If they do not perform in vivo studies, they need to modify their claims to indicate that the work was performed in silico and in vitro and further validation in vivo needs to be performed.

8) The authors need to provide more details in methods to make the study reproducible. This is critical. For example, the sequence for shRNAs needs to be provided. The methods for immunofluorescence staining is inadequate. It is critical to provide primary antibodies, suppliers, concentrations, lot numbers and all the details. It's not acceptable to write "experiments performed as previously described." Similarly more details for ChIP-Seq need to be provided and it cannot be written "ChIP-Seq performed as previously described". The primary antibodies for ChIP-Seq need to be provided. It's important for studies to be reproducible. All the lot numbers for all reagents and concentrations need to be provided as the methods are very thin on details. The code for bulk RNA-seq data analysis needs to be provided and the raw data deposited in a site such as GEO. It's critical that the authors provide all the code used for all the computational analysis in the paper and to deposit all the data in a site such as GEO. This will allow other researchers to use and benefit from the data, which isn't possible in the current form.

Overall this is an interesting study and I would be supportive of publication if the authors make the changes much of which revolve around making the data more transparent, code and raw data more accessible, and methods more detailed to improve reproducibility.

REVIEWER COMMENTS

Reviewer #1 (Remarks to the Author):

This study investigated the transcription dynamics of human limbal stem cells (LSCs) differentiation into corneal epithelial cells (CECs). By using the publicly available single cell RNA sequencing datasets, the authors successfully delineated a mechanism of RORA and PITX1 mediated histone post-translational modification, which further regulated the 3D chromosomal interactions to control the expression of CEC genes. This manuscript is well-written with advanced data analyses, supported by some validation cellular experiments. Here are my major comments:

1. Line 68-69: Please also provide some examples about what other cell types also express these markers (KLF4, PAX6, EHF and FOXC1).

Thank you for the reviewer's valuable suggestions. We add a sentence in the main text (line 67-70): "they are not CEC-specific and are also expressed in basal, suprabasal and superficial layers of the limbal epithelium." This can be demonstrated in the indicated citations.

2. Line 106: There are only 10 cell populations in Fig 1A, instead of 11 cell populations as stated in the main texts.

Thank you for the reviewer's valuable suggestions. We have corrected this error in the revised manuscript (line 107).

3. Please indicate the locations of limbus and central cornea in Fig 1D.

Thank you for the reviewer's valuable suggestions. we indicate the locations of the limbus and central cornea in Fig 1d in the revised version.

4. In Fig 1G, please also show the p-values of these GO terms.

Thank you for the reviewer's valuable suggestions. We add p-values of the GO terms in Fig 1g.

5. Line 136: About the statement of "canonical markers specific to the differentiating adult corneal epithelium and was involved in the response to cadmium, copper and zinc ions", please cite some references to support this claim.

Thank you for the reviewer's valuable suggestions. We have added two references (PAX6: Exp Eye Res, PMID:27818314 and J Cell Sci, PMID: 28202689) to support this claim in the revised manuscript (line 144).

6. Line 141: Why only focus on these 3 hub genes but not others like SPARC, GPC3, CRABP2, RPS3, RPL6, EEF1A1 and MEG3?

Thank you for the reviewer's valuable suggestions. We have listed all the hub genes (PTN, CRABP2, BCAM, GPC3, SPARC, MEG3, WNT6, EEF1A1, RPL6 and RPS3) in the revised manuscript (line 151-152).

7. Line 186: There are 21 TFs in Fig 2A, instead of 20 TFs as stated in the main texts.

Thank you for the reviewer's valuable suggestions. We have corrected this error in the revised manuscript (line 193).

8. Line 193: Please indicate RORA and PITX1 in Fig 2C.

Thank you for the reviewer's valuable suggestions. In order to show the gene expression changes better, we changed the volcano plot (Fig. 2c) to a heatmap and showed RORA and PITX1 in Fig 2c.

9. Again please indicate the locations of limbus and central cornea in Fig 2F.

Thank you for the reviewer's valuable suggestions. we indicate the locations of limbus and central cornea in Fig 2f in the revised version.

10. Line 261: It is very interesting to see proliferation, in addition to differentiation, was also affected in PITX1-OE and RORA-OE. The authors could discuss why cell proliferation is also affected.

Thank you for the reviewer's valuable suggestions. In discussion section of the revised manuscript, we discuss cellular proliferation as follows (line 429-434):

"It is well-known that cell cycle withdraw is required for stem cell differentiation. Interestingly, RORA and PITX1 also downregulated the proliferation-related genes like ETS1 and HMGA2, resulting in reduced LSC proliferation. Thus, the inhibition of LSC proliferation induced by RORA and PITX1 is important for LSC differentiation."

11: One minor comment on line 417: The word “underly” should be written as “underlie”.

Thank you for the reviewer’s valuable suggestions. We have corrected this spelling mistake in the revised manuscript (line 444).

Reviewer #2 (Remarks to the Author):

The manuscript presents an integrative study that combines a reanalysis of previously published single-cell data with new molecular biological techniques to explore differentiation in the corneal epithelium. This multi-faceted approach adds depth and breadth to the investigation and strengthens the study. However, there are some concerns in the manuscript (see specific points below):

1. Please discuss the relationship with previous reports in light of the current single-cell results, which have indicated that RUNX1, PAX6, and SMAD3 cooperatively interact with each other and establish core transcriptional regulatory circuits in limbal stem/progenitor cells.

Thank you for the reviewer’s valuable suggestions. In the discussion section, we discuss the relationship of RORA and PITX1 with RUNX1, PAX6 and SMAD3 as follows (line 434-442):

“Our previous study revealed that RUNX1, PAX6 and SMAD3 are expressed in both LSCs and CECs and play an indispensable role in corneal epithelial homeostasis⁴³. Loss of them not only disrupted CEC differentiation, but also induced cellular fate transition by activating the epidermal genes⁴³. In contrast, depletion of RORA and PITX1 only blocked CEC differentiation, but did not induce cellular fate switch. Thus, RUNX1, PAX6 and SMAD3 are required for corneal epithelial fate determination, while RORA–PITX1 axis is important to corneal epithelial differentiation.”

2. Previous report revealed that RORA strongly inhibited iPS induction from human corneal epithelial cells, indicating that RORA has a potential role in maintaining differentiated corneal epithelial cells (Cell Rep. 2016 May 10;15(6):1359-68. (PMID: 27134177)). It would be engaging for the authors to discuss it.

Thank you for the reviewer’s valuable suggestions. In the discussion section, we discuss it as follows (line 422-425):

“Interestingly, a previous report suggested that RORA strongly inhibits the induction of induced pluripotent stem cells from human CECs⁴², which further indicates that RORA plays a key role in maintaining CEC differentiation state.”

We also cite the paper (Cell Rep, 2016, PMID: 27134177) here.

3.It would be interesting to understand the cellular fate by showing what happens to cell proliferation-related or cell-cycle-related genes as the cells move toward the surface.

Thank you for the reviewer’s valuable suggestions. We show the gene expression patterns of the proliferation-related MYC, TP63, PEDF, and CEBPD along the developmental trajectory of the corneal epithelium in the UMAP plots (Supplementary Fig. 1e), as shown below. In the result section of the main text (line 122-127), we describe them as the following:

“We found that MYC²¹, TP63¹, PEDF²² and CEBPD⁷, which are associated with LSC proliferation, were highly expressed in adult LSCs and their expression dramatically decreased with the differentiation trajectory (Supplementary Fig. 1e). In addition, TP63 and PEDF were also expressed in embryonic LSCs (Supplementary Fig. 1e).”

Supplementary Fig. 1e

4. Introduction.

Authors should add the key references regarding transcription networks of corneal epithelial cells and limbal epithelial stem cells. (PAX6: Exp Eye Res. 2017 Jan;154:30-38 (PMID# 27818314), ABCB5: Nature. 2014 Jul 17;511(7509):353-7 (PMID: 25030174))

Thank you for the reviewer’s valuable suggestions. We have added these two references (line 144 and line 66) in the revised manuscript.

5. Line193-195: The chart does not indicate each gene's Log2FC values. This makes it difficult to assess the magnitude and significance of gene expression changes.

Thank you for the reviewer's valuable suggestions. In order to show the gene expression changes better, we changed the volcano plot (Fig. 2c) to a heatmap.

Fig. 2c

6. Line 646: While the t-test is commonly used for comparing two groups, it is not appropriate for multiple comparisons, as is the case in the current study.

Thank you for the reviewer's valuable suggestions. In our paper, t-test is used for comparing two groups (Fig.3d, 3k and Supplementary Fig. 3d, 3e). ANOVA was used for multiple comparisons (Fig. 3l). We have clarified them in the method section (line 699-700).

7. Figure 1C: It shows no difference in the expression of LSGALS3 in the PCW18 cell group compared to the other groups at other times a few weeks earlier than PCW18. Could the authors clarify the basis for their assertion that LSGALS3 is expressed in this group?

Thank you for the reviewer's valuable suggestions. Our assertion in the old manuscript is inaccurate. We rephrase it as follows (line 131-132):

“LGALS3 was primarily expressed in the suprabasal and superficial layers of the limbus and cornea from embryo to adulthood.”

8. Figure 2C: The volcano plots aim to present the differential gene expression data. However, it is unclear which plots correspond to which genes.

Thank you for the reviewer's valuable suggestions. In order to show the gene expression changes better, we changed the volcano plot (Fig. 2c) to a heatmap.

9. Figure 2J, 3H: The manuscript utilizes the kNN-DREMI score as both the vertical and horizontal axes. While the kNN-DREMI score is recognized for measuring relationships between genes, its use in this manner might not be appropriate.

Thank you for the reviewer's valuable suggestions. DREVI plots and DREMI values (Figures 2j, 3h) is common and robust to estimate gene-gene relationship, as shown in the cited paper (Cell, 2018, PMID: 29961576). In the old manuscript, due to our negligence, the DREMI values have been omitted in Fig. 2j and 3h. In the revised manuscript, we add the DREMI values to reveal the relationship between the indicated genes.

10. Figure 3E: The label on the vertical axis of the figure is marked as "ROR" when it appears that it should be labeled "RORA" to reflect the gene accurately.

Thank you for the reviewer's valuable suggestions. We have corrected this spelling mistake in the revised manuscript.

11. Figure 3L: The authors claim that OE significantly rescues the expression suppression caused by SR3335, but statistical evidence does not support this assertion.

Thank you for the reviewer's valuable suggestions. In the old version, the statistical results did not be indicated in Fig. 3l. In the revised manuscript, we performed statistical analysis and found that the gene expression changes were significant as shown in Fig. 3l.

Fig. 3l

12. Fig S4: If the target sequence for each transcription factor or enhancer region is known, please add it to the chart.

Thank you for the reviewer's valuable suggestions. The binding sequences namely motifs of each transcription factor have been shown in Supplementary Fig. 4b.

B Motif enrichment

TFs	motif	p-value
RORA		10^{-1494}
EHF		10^{-149}
ELF3		10^{-132}
PAX6		10^{-40}
KLF5		10^{-32}
KLF4		10^{-21}
PITX1		10^{-3}

Supplementary Fig. 4b

13. Fig 4H, 4I: Please add a description of the vertical scale. Chip-seq detects peaks, but authors should consider whether the positions of the peaks are consistent with the actual binding sequences of TFs.

Thank you for the reviewer's valuable suggestions. We add the vertical scale for each ChIP-seq track in Fig 4h, 4i and Supplementary Fig. 4e.

The well-known binding motif of each TF is the statistical result of a large amount of Chip-seq data. The binding of TFs to their target sequence depends on many factors like chromatin state, protein interaction partners, chromatin remodeling, histone epigenetic modifications etc. Thus, the appearance of a TF binding sequence does not reflect the actual TF binding. In addition, not all binding sites have the known TF binding sequence. TFs may bind to some sites that include an unknown binding sequence that has not been identified. ChIP-seq is a well-recognized and reliable approach to detect TF genome-wide binding sites. Therefore, Chip-seq peaks can and is sufficient to reveal the actual binding sites of TFs.

14. Figure 5: In the upper right corner of Figure 5, there appears to be a plot that suggests the representation of TADs. However, this part of the figure is not discussed in either the figure legend or the main text.

Thank you for the reviewer's suggestions. We are so sorry for the misunderstanding due to the lack of annotation in Figure 5. The image in the upper right corner of Figure 5 is a SCENIC TF regulon heatmap rather than a Hi-C map. We add annotations in figure 5 to indicate what they are.

In this paper, we only performed promoter capture Hi-C but not Hi-C as shown in Figure 4. Promoter capture Hi-C only generates a high-resolution map of the interactions between gene promoters and their regulatory elements. Our promoter capture Hi-C data can't obtain TADs.

Reviewer #3 (Remarks to the Author):

In the report, the authors performed analysis of previously published single cell RNA-seq data for development and adult cornea and identified RORA and PITX1 as key factors for promoting corneal epithelia cell (CEC) development from limbal stem cells (LCS). To test this prediction, over expression and knock down of RORA and PITX1 in primary LSC culture have been tested. Furthermore, CHIP-seq of RORA and PITX1 along with histone epigenetic marks and HiC experiments are used to investigate the mechanisms of gene regulation. The author's effort on experimentally testing the hypothesis is great and the results are very encouraging. The overall results are largely consistent with the informatic analysis from single cell data and the connection of RORA and PITX1 in CEC development is interesting. My specific comments are the following:

1. The authors reanalyzed previously published datasets. In figure 1A's UMAP, cells from PCW17 appear to segregate, forming the 'E_corneal superficial' cluster. This could be indicative of an uncorrected batch effect. Indeed, the RNA velocity analysis in figures 1I and 1K portrays potentially inconsistent trajectory to this cluster. Exploring alternative meta-analysis tools like scVI or Harmony might yield more coherent clustering.

Thank you for the reviewer's valuable suggestions. In Fig.1a, we used BBKNN (batch balanced k nearest neighbours) in scanpy to remove batch effect. As you see, our Fig.1a UMAP showed that the 'E_corneal superficial' cluster consisted mainly of PCW17, PCW10 and PCW21. But the "E_LSCs" cluster also contained a small number of PCW17 cells. We then used two other methods (CCA (canonical correlation analysis) + MNN (mutual nearest neighbor) and Harmony) to integrate these scRNA-seq data, as shown below. We found that these three methods yielded the same results. Fig.1c and Supplementary Fig.1i showed that CLU, LGALS3, MAL2, and S100A4 were expressed in 'E_corneal superficial' cluster but not in other embryonic cell clusters, suggesting that 'E_corneal superficial' cluster is a genuine cell cluster that is distinct from other embryonic cell types. So, we think this may be not a batch effect. PCW17 cells mainly segregate in 'E_corneal superficial', probably because that the corneal superficial cells were the major cell type captured in PCW17 sample.

In Fig.1 k (in the old manuscript), dynamo builds a cell-wise transition matrix by translating the velocity vector direction and the spatial relationship of each cell to its neighbors to transition probabilities. Dynamano implements such a functionality (`dynamano.pandas.state_graph`) to effectively creates a model that summarizes the possible cell type transitions based on the reconstructed Markov transition matrix between cell or the vector field function. This builds a transition graph between different cell types (Fig. 1k in the old manuscript). Thus, Fig. 1k (in the old manuscript) was generated in a method that is quite different from RNA velocity. RNA velocity can only predict the developmental flow between adjacent cells, while dynamano transition graph can infer the developmental relationship of each cell type with all other cell types. Although the cell-state transition between other cell types recapitulated the natural development and differentiation trajectory, the `dynamano.pandas.state_graph` function failed to infer the developmental trajectory of “E_corneal superficial”, as shown in fig. 1k (in the old manuscript). As a bioinformatics tool, it is difficult to make accurate predictions in all aspects of a complex system. Thus, we delete Fig.1k (in the old manuscript) in the revised manuscript.

2. RORA and PITX1 don't exhibit high expression until later developmental stages and are predominantly expressed in terminally differentiated CECs. This would lead one to anticipate that these factors play roles in CEC terminal differentiation rather than as precursory factors transitioning LSC to CEC. How do the authors reconcile this seeming discrepancy? Is there any evidence of these genes being active in proliferating progenitor cells derived from LSC in the adult cornea?

Thank you for the reviewer's valuable suggestions. LSCs asymmetrically divide and produce both stem cells and transit-amplifying cells (TACs). TACs further proliferate, migrate upward and centripetally, and eventually differentiate into three specialized post-mitotic corneal epithelium cell types: basal, suprabasal and superficial cells. TACs is a proliferating progenitor cell population derived from LSCs in the adult cornea. As there are very few TACs in the corneal

epithelium, no TACs were detected in the scRNA-seq data used in this paper. We downloaded and re-analyzed another scRNA-seq data from GSE153515 (*The Ocular Surface*, 2021, PMID: 33388438), identifying a TAC cluster. We found that the TAC population expressed the proliferation-associated TAC markers MKI67, BIRC5 and TOP2A. We also found that RORA was expressed in TACs, as shown below. Thus, RORA was activated in the very early stages of differentiation. In addition, in our work, overexpression of RORA in LSCs inhibited LSC proliferation and directedly induced LSCs to differentiation into CECs in the absence of differentiation media (Figure 3). Therefore, we think that RORA was able to initial and drive CEC terminal differentiation.

3. While the overexpression of RORA and PITX1 clearly induces marker gene expression characteristic of CECs and decrease in LSC proliferation in culture. Given their roles as transcription factors, it's not unexpected for their overexpression to instigate a cascade of gene expression. Yet, it's crucial to establish if their overexpression truly facilitates the LSC to CEC developmental transition. Validating that cultured cells undergo accelerated CEC development and result in genuine CECs is essential.

Thank you for the reviewer's valuable suggestions. The keratin proteins KRT3 and KRT12 were the most canonical and loyal markers to identify mature CECs. They are also used to distinguish the corneal epithelium from other epithelia. Unfortunately, other than the markers, there is no other method used to identify corneal epithelium until now. In our paper, overexpression of RORA activated CEC markers including KRT3, KRT12, LGALS3, KRT24 and CLU. RORA also upregulated the transcription factors PAX6 and KLF4, which are well-known to promote CEC differentiation. In addition, GSEA (Fig.3f, g) showed that those in-vivo-CEC-specific and in-vitro-induced-CEC-specific gene sets were significantly up-regulated overall, while in-vivo-LSC-specific and in-vitro-LSC-specific gene sets were down-regulated overall. Conversely, loss of RORA inhibited the expression of the CEC-specific gene sets upon differentiation (Supplementary Fig. 3f). Importantly, our results suggested that RORA overexpression established the CEC-specific chromatin epigenetic landscape that contributed to the activation of the CEC gene sets (Figure 4 and Supplementary Figure 4). These evidences suggest that RORA-induced cells are genuine CECs.

Fig. 3f

Fig. 3g

Supplementary Fig. 3f

4. Given the inherent differences between in vitro and in vivo settings, corroborating the central hypothesis using animal models like mice or zebrafish, especially through gene knockouts or knockdowns during development, would be critical.

Thank you for the reviewer's suggestions. As shown in the following images, we found that RORA was not expressed in the murine corneal epithelium. We used five anti-RORA antibodies from different suppliers and RORA was not detected in the murine corneal epithelium. As we all know, the mouse corneal epithelium is quite different from that of humans in terms of gene expression. So the animal models like RORA-knockout mice can't be used to explore the *in vivo* function of RORA.

Reviewer #4 (Remarks to the Author):

This is an interesting study where the authors use a single-cell transcriptomic atlas to show the molecular dynamics during human corneal epithelial cell development and differentiation. They show the role of RORA and PITX1, primarily in silico in driving limbal stem/progenitor cells to differentiate into corneal epithelial cells. Overall the study is of interest and well-performed. Comments are below.

1) At the low level of resolution and image quality, it is challenging to determine co-localization of the limbal stem cell maker S100A2 and KRT15. The authors need to provide higher quality images at greater magnification. It is also necessary to perform quantification showing the degree of overlap between S100A2 and KRT15.

Thank you for the reviewer's valuable suggestions. We provide higher quality IF images in the revised manuscript. In the result section, we describe it as follows (line 119-120):

“S100A2 was located at the basal layer of the limbus and more than half of the S100A2-positive cells were KRT15-positive.”

2) The authors claim that limbal stem cells express IFITM3; however, in the right panel there is no co-localization with a marker. The authors need to perform double immunofluorescence with a limbal stem cell marker to make this claim.

Thank you for the reviewer's valuable suggestions. A previous study (*Cell Stem Cell*, 2021, PMID: 33984282) showed that IFITM3 was a LSC marker and supported undifferentiated state of LSCs. We performed co-staining of IFITM3

and KRT19 (a canonical LSC marker) and found that IFITM3 was co-located with KRT19 in the basal layer of the limbus (Fig. 1d) (line 120-122).

Fig. 1d

3) The authors should refrain from using short-hand terms. For example, the text states that the limbal stem cell markers KRT15 and KRT14 were used (line 115), whereas the figure states K12 and K15. It's confusing why the text refers to KRT14; however, the figure states K12? The authors should clarify this. There is no notation in the figure legend or text that K12 or K15 refer to KRT15 and KRT14. Thus the authors need to write out all gene names in full. The authors should also write out corneal epithelial cells and limbal stem/progenitor cells each time rather than CECs for people outside the field who are not familiar with these terms. Similarly they should write out overexpression rather than the term OE.

Thank you for the reviewer's valuable suggestions. In the revised manuscript, we use the gene full name KRT12, KRT15, KRT14 in the main text and all figures. We also changed OE to overexpression. However, we keep the abbreviations LSCs and CECs. As you see, the full names of these two abbreviations are too long in the figures. Our figure layout is very full and lacks extra space. If we used the long full names in the figures, it will seriously affect the layout and aesthetics of the figures.

4) In the text the authors state that LSGALS3 was primarily expressed in the suprabasal and superficial layers of the limbus and cornea; however, in the figure uses the term LGALS3. I assume the authors are referring to LGALS3, and this appears to be a mistake in the text, which should be corrected. The authors need to provide a higher resolution image in Figure 1D adult for LGALS3, as it's not possible to make out the expression in the suprabasal and superficial layers at low resolution.

Thank you for the reviewer's valuable suggestions. We have corrected this spelling mistake in the revised manuscript. We also provide a higher resolution Figure 1d.

5) The authors need to annotate all the figures of the corneas to show where the suprabasal, superficial, and limbal-corneal regions are. The is important to

perform as we cannot tell the regions or corneal histology based on DAPI nuclear staining alone.

Thank you for the reviewer's valuable suggestions. We provide a H&E staining in Supplementary Fig. 1a with a great magnification to indicate the basal, suprabasal, and superficial layers of the limbal-corneal epithelial tissue.

Supplementary Fig. 1a

6) The authors state that PITX1 is expressed in the differentiated limbal-corneal epithelial layer of adults; however, no co-staining is performed. This should be performed with markers. Similarly we cannot determine if there is co-localization of RORA and KRT3 at the low level magnification. Quantification should be performed on co-localization studies.

Thank you for the reviewer's valuable suggestions. We perform co-staining of PITX1 and KRT3 in Fig. 2f. The differentiated limbal epithelium includes the suprabasal and superficial layers, while the differentiated central corneal epithelium includes the basal, suprabasal and superficial layers. KRT3 is primarily expressed in the suprabasal and superficial layers of the central corneal epithelium. In Fig. 2f, KRT3 was only used to distinguish central cornea from limbus. As shown in Fig. 2f, almost all of the differentiated limbal-corneal epithelial layer showed the expression of RORA and PITX1.

Fig. 2f

7) The authors make the claim "In a word, we highlighted that RORA-PITX1 axis is necessary and sufficient to dictate corneal epithelial differentiation." Most of the studies are performed in silico with the exception of in vitro studies of limbal stem cells. The author need to perform conditional knockout in mouse of RORA-PITX1 to make the claim that the RORA-PITX1 axis is necessary and sufficient to dictate corneal epithelial differentiation. If they do not perform in

vivo studies, they need to modify their claims to indicate that the work was performed in silico and in vitro and further validation in vivo needs to be performed.

Thank you for the reviewer's valuable suggestions. We rewrite the description in the main text (line 299-300) as follows:

"In a word, we highlighted that RORA-PITX1 axis plays a key role in the corneal epithelial differentiation *in vitro*."

8) The authors need to provide more details in methods to make the study reproducible. This is critical. For example, the sequence for shRNAs needs to be provided. The methods for immunofluorescence staining is inadequate. It is critical to provide primary antibodies, suppliers, concentrations, lot numbers and all the details. It's not acceptable to write "experiments performed as previously described." Similarly more details for ChIP-Seq needs to be provided and it cannot be written "ChIP-Seq performed as previously described". The primary antibodies for ChIP-Seq need to be provided. It's important for studies to be reproducible. All the lot numbers for all reagents and concentrations need to be provided as the methods are very thin on details. The code for bulk RNA-seq data analysis needs to be provided and the raw data deposited in a site such as GEO. It's critical that the authors provide all the code used for all the computational analysis in the paper and to deposit all the data in a site such a GEO. This will allow other researchers to use and benefit from the data, which isn't possible in the current form.

Overall this is an interesting study and I would be supportive of publication if the authors make the changes much of which revolve around making the data more transparent, code and raw data more accessible, and methods more detailed to improve reproducibility.

Thank you for the reviewer's valuable suggestions. we have changed the method section as required by the reviewer.

The shRNA sequences for *PITX1* are provided in the method section (line 491-492).

We provide detailed information of primary antibody that were used in immunofluorescence staining (line 522-529) and ChIP-Seq (line 562-566).

We provide a more detailed ChIP-Seq method.

We provide a detailed information for all reagents used in this paper.

we add a "code availability" statement as follows (line 712-714):

"The code of bioinformatics analysis in this paper is available through GitHub (<https://github.com/Mingsenli/corneal-single-cells>)"

In the “Data availability” section, we state as follows (line 703-710):

“All the raw sequence data generated in this paper have been deposited in the Genome Sequence Archive in National Genomics Data Center, China National Center for Bioinformation (GSA-Human: HRA005797) that are publicly accessible at <https://ngdc.cncb.ac.cn/gsa-human>.”

Note: As soon as this paper is online, the raw data can be downloaded from the deposited database.

REVIEWERS' COMMENTS

Reviewer #1 (Remarks to the Author):

The authors have addressed all my previous comments. I have no further comment.

Reviewer #2 (Remarks to the Author):

Authors have provided a detailed point-by-point response to comments, addressed issues raised by the reviewer.

Reviewer #3 (Remarks to the Author):

The authors had done a great job addressing issues raised by the reviewers. I believe the revised manuscript is ready to be accepted for publication.

Reviewer #4 (Remarks to the Author):

Overall, the authors have addressed my comments and I am now supportive of publication.